# Major Histocompatibility Complex Class I Chain-Related α (MICA) STR Polymorphisms in COVID-19 Patients

**DOI:** 10.3390/ijms23136979

**Published:** 2022-06-23

**Authors:** Juan Francisco Gutiérrez-Bautista, Alba Martinez-Chamorro, Antonio Rodriguez-Nicolas, Antonio Rosales-Castillo, Pilar Jiménez, Per Anderson, Miguel Ángel López-Ruz, Miguel Ángel López-Nevot, Francisco Ruiz-Cabello

**Affiliations:** 1Servicio de Análisis Clínicos e Inmunología, Hospital Universitario Virgen de las Nieves, 18012 Granada, Spain; juanfry@correo.ugr.es (J.F.G.-B.); antoniorn87@gmail.com (A.R.-N.); mpilar.jimenez.sspa@juntadeandalucia.es (P.J.); per.anderson@ibsgranada.es (P.A.); 2Programa de Doctorado en Biomedicina, University of Granada, 18016 Granada, Spain; 3Departamento Bioquímica, Biología Molecular e Inmunología III, University of Granada, 18071 Granada, Spain; 4Servicio de Análisis Clínicos, Hospital Universitario de Jaén, 23007 Jaén, Spain; alba.martinez.sspa@juntadeandalucia.es; 5Servicio de Medicina Interna, Hospital Universitario Virgen de las Nieves, 18012 Granada, Spain; antonio.rosales.castillo.sspa@juntadeandalucia.es; 6Instituto de Investigación Biosanitaria de Granada (ibs.GRANADA), 18012 Granada, Spain; 7Departamento de Medicina, University of Granada, 18071 Granada, Spain; mangel.lopez.sspa@juntadeandalucia.es; 8Servicio de Enfermedades Infecciosas, Hospital Universitario Virgen de las Nieves, 18012 Granada, Spain

**Keywords:** MICA STR polymorphisms, MICA, SARS-CoV-2, NK cells, innate immunity

## Abstract

The SARS-CoV-2 disease presents different phenotypes of severity. Comorbidities, age, and being overweight are well established risk factors for severe disease. However, innate immunity plays a key role in the early control of viral infections and may condition the gravity of COVID-19. Natural Killer (NK) cells are part of innate immunity and are important in the control of virus infection by killing infected cells and participating in the development of adaptive immunity. Therefore, we studied the short tandem repeat (STR) transmembrane polymorphisms of the major histocompatibility complex class I chain-related A (MICA), an NKG2D ligand that induces activation of NK cells, among other cells. We compared the alleles and genotypes of MICA in COVID-19 patients versus healthy controls and analyzed their relation to disease severity. Our results indicate that the MICA*A9 allele is related to infection as well as to symptomatic disease but not to severe disease. The MICA*A9 allele may be a risk factor for SARS-CoV-2 infection and symptomatic disease.

## 1. Introduction

The SARS-CoV-2 infection induces an early immune response in which various elements of innate immunity participate. Natural killer (NK) cells represent an important component of innate immunity participating in the defense against viral infections through their (i) production of perforin and granzymes, which eliminate infected cells [1]; (ii) induction of apoptosis through caspase-dependent and independent pathways [1]; (iii) promotion of the innate and adaptive responses through the production of cytokines and chemokines [2]; (iv) production of gamma interferon (IFN-γ) that stimulates cytotoxic CD8+ T cells and favors the differentiation of CD4+ T cells into T-helper (Th) 1 cells, which are important in viral infections [3]; and (v) production of antibody-dependent cytotoxicity via CD16 [1].

NK cells express a number of inhibitory and activating receptors that recognize various molecules, including major histocompatibility complex class I (MHC-I), that promote their antiviral capacities, while maintaining self-tolerance [4,5]. The NKG2D receptor is a potent activator of NK cells [6], which is also present on CD4+ T cells, CD8+ αβ T cells, γδ T cells, and iNKT cells [7]. NKG2D can directly induce the cytotoxic function and IFN-γ production of NK cells, while it acts as a costimulator of the cytotoxic activity in CD8+ αβ T cells [6,7].

The major histocompatibility complex class I chain-related A (MICA) is one of the NKG2D ligands [8]. These are highly polymorphic nonclassical MHC proteins, of which 108 alleles have been detected [7]. MICA is expressed on the cell surface of the epithelial cells, fibroblasts, keratinocytes, endothelial cells, and monocytes under stress conditions, such as a virus infection [1,9]. The structure of MICA is similar to that of MHC-I molecules, but it is not associated with β2-microglobulin [8]. It has three extracellular domains (α1, α2, and α3) encoded by exons 2, 3, and 4, respectively, together with a transmembrane region encoded by exon 5 [8]. This transmembrane region has a short tandem repeat (STR) of GCT triplets, which codify alanine residues [8]. The different number of GTC repetitions (4, 5, 6, 7, 8, 9, or 10) gives rise to the alleles A4, A5, A6, A7, A8, A9, and A10, respectively [10]. In addition, there is an extra allele (A5.1) that has an insertion of a G in exon 5, which produces a stop codon and therefore a truncated protein that does not have the transmembrane region [10]. The transmembrane region is involved in the physiological localization of the MICA molecule. The alleles that have intact transmembrane regions present a basolateral expression, while the A5.1 allele presents an apical localization [11]. The A5.1 allele is expressed on the membrane via a glycosyl-phosphatidyl-inositol (GPI) anchor [12], with the highest levels of soluble MICA (sMICA) found in carriers of this allele [13].

Decreased NK cells numbers and their reduced ability to kill infected cells have been reported in coronavirus disease 2019 (COVID-19) patients [14]. The ability of the NK cell to bind to the target cell is altered, reducing its cytotoxic capacity [14]. Therefore, the presence of certain MICA alleles may favor NK cell cytotoxicity and allow better control of the disease. In contrast, other alleles can be related to a worse prognosis, as evidenced by a study linking an increase in sMICA with symptomatic COVID-19 [15]. Thus, allelic and biomechanical differences of MICA may be important factors for its interaction with NKG2D, thus modulating the function of NK cells and the innate immune response.

Previously, we have performed a study of the classic MHC molecules and their relationship with COVID-19 [16]. Currently, we have studied the association of COVID-19 with STR in exon 5 of the MICA gene, HLA-B genes, and HLA-B/MICA haplotypes and in asymptomatic, moderate, and severe COVID-19 patients. In addition, we have compared the presence of these alleles with biochemical parameters, days of hospital admission, and other factors, attempting to clarify the role of MICA in COVID-19 infection.

Our hypothesis is that MICA molecules are very important for the activation of NK and T cells via NKG2D in infectious processes, such as COVID-19. The different MICA alleles may vary in the activation of these cells and produce different degrees of activation and control of the infection.

## 2. Results

In the study of STR polymorphisms of MICA, the alleles A4, A5, A5.1, A6, and A9 were detected (Table 1). The allele with the highest representation in the population reference group and COVID-19 patients was MICA*A6, while the one with the lowest frequency was MICA*A5. The comparison of the allelic and genotypic frequencies between COVID-19 cases and the population reference group are presented in Table 2. However, we made comparisons between the different COVID-19 patients based on disease severity, grouped as asymptomatic, symptomatic, moderate, and severe patients, with each other and with the population reference group. 

The result showed a statistical difference with a higher allele frequency of MICA*A9 in COVID-19 patients versus the population reference group (*p* = 0.004, Pc = 0.025, odds ratio = 1.399, 95% confidence interval = 1.110–1.762) (Table 1). When we compared asymptomatic and symptomatic patients versus the population reference group, only the symptomatic patients had a statistically higher frequency of the MICA*A9 allele. In addition, these differences were maintained when the comparison was performed between moderate patients and the population reference group. The rest of the comparisons among the different groups did not show a significant value.

The genotype comparisons did not show any statistical difference among the groups (Table 1). The genotype MICA*A9 homozygous showed a higher frequency in moderate patients versus the population reference group (*p* = 0.041), but the significance was lost after Bonferroni correction (Table 1).

The association between MICA and coinfection or sepsis development in COVID-19 patients was not possible to study due to the low number of cases in our studied population.

Finally, due to the age difference between cases and the population reference group, we studied a possible association between age and MICA STR alleles. We did a regression model that did not show any relation between age and MICA STR alleles (R Square “R2” = 0.004).

Additionally, we made the comparison between homozygous versus heterozygous genotypes in all COVID-19 groups, with each other and versus the population reference group, but did not find significant results (data not shown).

Due to the proximity of the HLA-B and MICA loci, we studied the HLA-B frequency and the HLA-B/MICA haplotypes.

The frequencies of the HLA-B alleles are presented in Appendix A. Alleles with a frequency lower than 1% were excluded from the study. There were no statistical differences between the groups and the population reference group. Only HLA-B*50 showed a higher frequency (*p* = 0.027, Pc= n.s) in severe (4.9%) compared to moderate patients (2%).

We next carried out the construction of the HLA-B/MICA haplotypes, studying those that presented a frequency > 1% (Table 3). The comparisons showed that the HLA-B*57/MICA*A9 haplotype was more frequent in COVID-19 cases than in the population reference group but significance was lost upon Bonferroni correction. We then analyzed the MICA*A9 haplotypes together obtaining a statistical difference (Pc = 0.01) between COVID-19 patients versus the population reference group. This result was due to MICA*A9. In the intragroup comparison, the HLA-B*50/MICA*A6 haplotype was more represented in severe (4.9%) than in moderate (2%) or asymptomatic patients, although significance was lost after Bonferroni correction.

Finally, we compared HLA-B and MICA with different biochemical parameters (lactate dehydrogenase, IL-6, platelets, PO2, bilirubin, creatinine, D-dimer, C reactive protein, ferritin, troponin, natriuretic peptide, procalcitonin, and fibrinogen), days of hospital stay or stay in the ICU, age at diagnosis, and the need for respiratory support (conventional oxygen therapy, noninvasive ventilation, or intubation). We did not detect any significant associations among the above parameters, HLA-B, MICA, or the HLA*B/MICA haplotypes (data not shown).

## 3. Discussion

The SARS-CoV-2 infection has shown different response phenotypes (asymptomatic, moderate and severe disease, or death). These differences among individuals may be due to innate immunity. Therefore, we have tried to identify a possible role of NK cells by studying the different MICA alleles. NK cells may present variations in their response to MICA due to polymorphisms in its transmembrane region [17]. STR polymorphisms in the transmembrane region have been related to susceptibility or protection to various infectious agents [10,18].

We found a higher frequency of MICA*A9 in COVID-19 patients versus PRG. In addition, MICA*A9 was higher in symptomatic versus asymptomatic patients, with the highest frequency found in patients with moderate disease severity. Moreover, we did not find significance in severe patients due to the small population studied. MICA*A9 associations is independent from HLA-B locus.

The extracellular domain of MICA protein has a polymorphism at the amino acid position 129 that is important for the NKG2D interaction. The variant MICA V129 (valine in 129) is associated with reduced affinity for NKG2D, whereas M129 (methionine in 129) confers high-binding affinity to NKG2D [19] (Figure 1). All MICA*A9 alleles shared M129, forming a haplogroup, presenting a high affinity for the NKG2D receptor [20].

SARS-CoV-2 infection generates more sMICA by ADAM17 metalloproteinase overexpression after ACE2 and spike protein interaction that decrease the activation of NK cells [21] (Figure 1). This immune evasion will be more effective in individuals with MICA alleles with high affinity for NKG2D, such as MICA*A9, inducing a stronger inhibition to NK and T cells and allowing an increased replication and dispersion of the virus.

Castelli et al. reported an association of MICA rs2596541 variants with symptomatic patients [15]. These variants increase mRNA levels and may favor the increase in sMICA. Furthermore, they observed that MICA*008 and MICA*019, which are in linkage disequilibrium with rs2596541, were increased in symptomatic patients [15]. The MICA*008 allele is associated with the STR MICA*A5.1 polymorphism, while the MICA*019 is associated with MICA*A5 [22]. Our results did not show significant results associated with symptoms or severity in any of these STR polymorphisms. However, it should be noted that each of the STR polymorphisms has more than one associated MICA allele [22].

Otherwise, the MICA expression level might be related to host MICA gene polymorphisms [23]. Nevertheless, virus infections can interfere with MICA expression. For example, in HCMV infection some MICA alleles are downregulated while MICA*008 is not. MICA*008 is in linkage disequilibrium with MICA*A5.1. Therefore, the cytotoxic effects of NK cells against HCMV-infected cells were stronger in individuals carrying MICA*008/MICA*A5.1 [23]. However, UL142-HCMV protein downregulates the expression of MICA proteins that have a large transmembrane region but not those with a short transmembrane region [12]. Moreover, SARS-CoV-2 acts in a similar manner, which makes individuals with MICA*A9 more at risk of infection and symptomatic diseases.

The allele MICA*A9 has been associated with a diversity of diseases. A lower frequency of MICA*A9 has been reported in syncytial virus infection (RSV) than in controls [19]. The presence of MICA*A9 in heterozygosity can protect against Chlamydia trachomatis infection [20]. Furthermore, MICA*A9 has a relation with autoimmune disease [21,22,23,24].

The studies in COVID-19 patients showed increased levels of exhausted NK cells. These NK cells exhibit a high level of programmed death 1 (PD-1) and NK group 2 member A receptor (NKG2A) expression [24,25]. NKG2A is one of the most important inhibitor-receptors of NK cells [26]. Also, other NK cells polymorphisms are related to COVID-19. The killer cell immunoglobulin-like receptors (KIRs) are a family of highly polymorphic transmembrane glycoproteins that induced inhibitory or activating signals to NK and T cells upon recognition of their ligands [27]. Studies showed that KIR2DS4 is related to severe COVID-19 infection [28,29].

MICA, similar to the other mentioned factors, is not only important for the correct function of NK cells but also for adaptative immunity. NK cells intervene in adaptive immunity due to their production of interferon-γ (IFN- γ) and IL-2, which induce the activation of CD4+ T cells and favor their differentiation into Th1 cells [30,31]. This is one reason for the low Th1 levels of COVID-19 patients [3].

It is known that the MICA-NKG2D interaction is important for the activation of NK and T cells for the elimination of cells stressed by tumors or infections [32]. Recent studies in the area of cancer have directed efforts to create a vaccine that prevents the production of sMICA due to its role in the immune escape, producing the inhibition of NK and T cells [33].

In conclusion, this study tries to better explain the different factors that model the NK cells and their impact on SARS-CoV-2 infection. The identification of MICA polymorphisms and susceptibilities to different diseases could promote earlier diagnosis and preventive measures. Our results indicate that the STR polymorphisms in MICA*A9 has an impact on the risk of contagion and disease severity. MICA*A9 is a possible factor of innate immunity that could help explain the risk of infection and the different responses observed in SARS-CoV-2-infected individuals. However, although there seems to be no relationship between age and the frequency of MICA STR alleles, a possible limitation of the study is the age difference between the population reference group and the cases. Therefore, our conclusions need to be further verified in other populations.

## 4. Materials and Methods

### 4.1. Samples

The study population consisted of 483 individuals diagnosed with COVID-19 at the University Hospital Virgen de las Nieves. All the individuals tested positive in polymerase chain reaction (PCR) for SARS-CoV-2. These individuals were classified according to the severity of the disease:

Asymptomatic: 33 patients (23 women and 10 men); mean age of 42.6 years (26–63).Symptomatic:a.Moderate patients: 344 patients (158 women and 186 men); mean age of 63.6 years (25–98).b.Severe patients: 106 patients (38 women and 67 men); mean age of 61.8 years (26–86).


The severity of the disease was determined following the guidelines of the National Institutes of Health [33]:(a)Asymptomatic: Individuals who test positive for SARS-CoV-2 using a virologic test but who have no symptoms that are consistent with COVID-19 [34].(b)Moderate Illness: Individuals who show evidence of lower respiratory disease during clinical assessment or imaging and who have an oxygen saturation (SpO_2_) ≥ 94% on room air at sea level [34].(c)Severe Illness: Individuals who have SpO_2_ < 94% on room air at sea level, a ratio of arterial partial pressure of oxygen to fraction of inspired oxygen (PaO_2_/FiO_2_) < 300 mm Hg, a respiratory rate > 30 breaths/min, or lung infiltrates > 50% [34].

The patients were recruited between April 2020 and January 2021. The variants of SARS-CoV-2 prevalent at this time in Spain were Wuhan and Alfa [35]. In addition, there were no vaccines available at this time.

The comorbidities and characteristics of the study population are summarized in Table 3.

All patient samples were collected according to the local medical ethics regulation after informed consent was obtained by the subjects, their legal representatives, or both, according to the Declaration of Helsinki. The studies involving human participants were reviewed and approved by the ethics committee via the Portal de Ética de la Investigación Biomédica de Andalucía (PIEBA) of the Andalusian Government (Code: 0766-N-20).

### 4.2. Population Reference Group

The population reference group (PRG) consisted of 617 blood samples from donors from the regional blood bank of Granada. The PRG was recruited from 2005 to 2015. Healthy individuals who did not present any disease of autoimmune etiology or immunodeficiencies were selected. The average age of the group is 45 years and 325 (51%) are women. This group was used as a representation of the allele and genotype frequencies in our population.

### 4.3. DNA Extraction

DNA was extracted from peripheral blood of each of the patients using the QIAMP DNA Blood Mini Kit (Qiagen, Hilden, Germany) according to the manufacturer’s instructions.

### 4.4. HLA-B and MICA Genotyping

HLA-B typing was performed at high resolution using the LABType sequence-specific oligonucleotide typing test (One Lambda, Canoga Park, CA, USA). Target DNA was amplified by PCR using sequence-specific primers, followed by hybridization with allele-specific oligodeoxynucleotides coupled with fluorescent phycoerythrin-labeled microspheres. Fluorescence intensity was determined using a LABScan 100 system (Luminex xMAP, Austin, TX, USA). HLA alleles were assigned using the HLA-Fusion software (version 4.6.013925) (One Lambda).

The study of the STR polymorphism of exon 5 of MICA was performed by polymerase chain reaction (PCR) following the method of Ota et al. (https://pubmed.ncbi.nlm.nih.gov/9174136/) (accessed on 10 February 2022).

The amplified products were analyzed by electrophoretic separation using the ABI Prism 3130XL Genetic Analyzer (Applied Biosystems, Foster City, CA, USA), and their sizes were analyzed using the GenMapper version 4.0 software (Applied Biosystems).

### 4.5. PCR Diagnosis of SARS-CoV-2 Infection

PCR diagnosis was performed using the cobas^®^ SARS-CoV-2 assay on a cobas^®^ 6800 system (Roche Molecular Systems, Pleasanton, CA, USA). This is a single-well, double-target assay that enables both the specific detection of SARS-CoV-2 and the detection of pan-Sarbecovirus of the Sarbecovirus subgenus family, which includes SARS-CoV-2. The test detects the genetic signature (RNA) of the SARS-CoV-2 virus in nasal, nasopharyngeal, and oropharyngeal swab samples from patients who meet COVID-19 clinical and/or epidemiological criteria for testing.

### 4.6. Statistical Analysis

Statistical analysis was performed to compare allelic, genotypic, and haplotypic distributions among patients and PRG using the X2 test or the two-tailed Fisher’s exact test, when necessary, with contingency tables. Significance levels were corrected by Bonferroni correction for a multiplicity of testing by the number of comparisons. The risk estimation was determined by calculating the odds ratio (OR) with a confidence interval (CI) of 95%. The Kruskal–Wallis and U Mann–Whitney were used to compare groups when the distribution was not normal (as checked by the Kolmogorov–Smirnov test). The software used was SPSS statistical software (Windows version 26, IBM, Armonk, NY, USA).

We applied a lineal regression using R Square “R2” to estimate the association between age and MICA STR allele.

The estimation of haplotype frequencies, LD, and haplotypically based hypothesis tests were calculated using Arlequin version 3.1 (Appendix A), the software package for population genetics.

A corrected *p*-value of <0.05 was considered statistically significant.

## Figures and Tables

**Figure 1 ijms-23-06979-f001:**
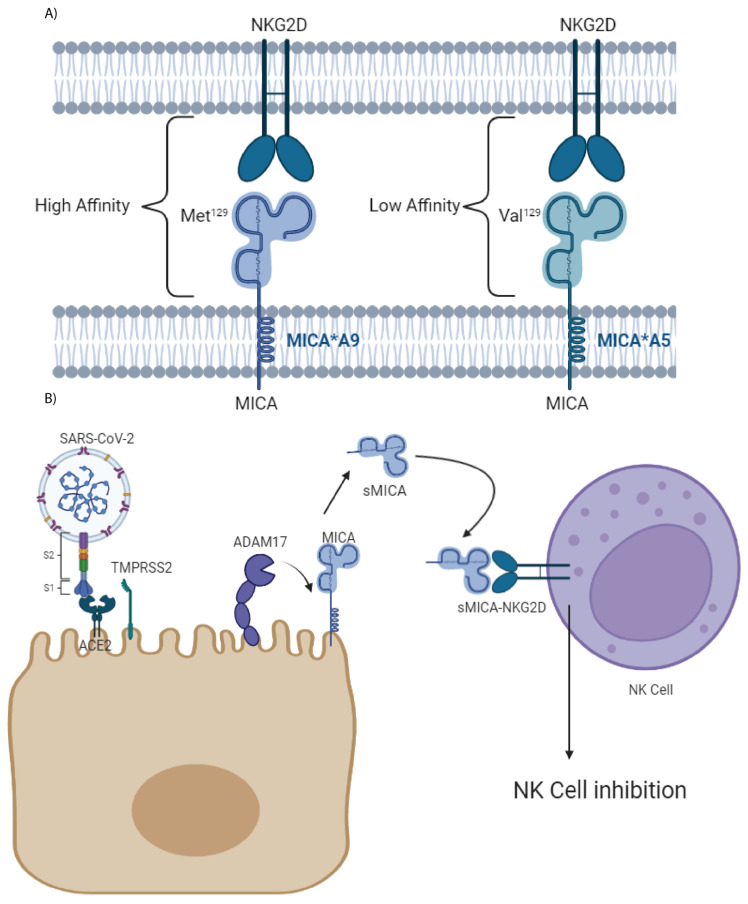
Graphic representation of the MICA-NKG2D interaction depending on the Met/Val polymorphism at 129 and the production of sMICA by ADAM17: (**A**) The Met/Val polymorphism at 129 of the MICA molecules influences binding to NKG2D. Alleles with Met^129^ have high affinity for the ligand, while those with Val^129^ have low affinity. The MICA*A4 and *A9 alleles have Met^129^, MICA*A5 present Val^129^, and the MICA*5.1 and *A6 alleles can have both polymorphisms [20]. (**B**) In COVID-19 infection, the metalloprotease ADAM17 hydrolyzes MICA generating sMICA, which binds to NKG2D causing inhibition. In the case of the MICA alleles with the Met129 variant, the inhibition will be greater, and there will be a worse control of the infection by the NK and T cells. Abbreviations—ACE2: angiotensin-converting enzyme 2; S1: S1 subunit of spike protein; S2: S2 subunit of spike protein; TMPRSS2: transmembrane protease serine 2; ADAM17: ADAM metallopeptidase domain 17.

**Table 1 ijms-23-06979-t001:** MICA alleles and genotypic frequencies.

Allele	PRG (2n = 1234) n (%)	COVID-19 Patients (2n = 892) n (%)	Asymptomatic Patients (2n = 66) n (%)	Moderate Patients (2n = 686) n (%)	Severe Patients (2n = 206) n (%)	P1 (Pc)	P2 (Pc)	P3 (Pc)	P4 (Pc)	P5 (Pc)	P6 (Pc)
*MICA*A4*	192 (15.6)	119 (13.3)	11 (16.7)	95 (13.8)	24 (11.7)	n.s	n.s	n.s	n.s	n.s	n.s
*MICA*A5*	140 (11.3)	102 (11.4)	6 (9.1)	81 (11.8)	21 (10.2)	n.s	n.s	n.s	n.s	n.s	n.s
*MICA*A5.1*	310 (25.1)	196 (22.0)	19 (28.8)	158 (23)	38 (18.4)	n.s	n.s	n.s	n.s	n.s	n.s
*MICA*A6*	423 (34.3)	309 (34.6)	22 (33.3)	226 (32.9)	83 (40.3)	n.s	n.s	n.s	n.s	n.s	n.s
*MICA*A9*	169 (13.7)	166 (18.6)	8 (12.1)	126 (18.4)	40 (19.4)	0.004 (0.025)	n.s	0.007 (0.035)	n.s	n.s	n.s
**Genotype**	**PRG (*n* = 617) n (%)**	**COVID-19 Patients (*n* = 446) n (%)**	**Asymptomatic Patients (*n* = 33) n (%)**	**Moderate Patients (*n* = 343) n (%)**	**Severe Patients (*n* = 103) n (%)**	**P1 (Pc)**	**P2 (Pc)**	**P3 (Pc)**	**P4 (Pc)**	**P5 (Pc)**	**P6 (Pc)**
*MICA*A4,*A4*	12 (1.9)	4 (0.9)	1 (3)	4 (1.2)	0 (0)	n.s	n.s	n.s	n.s	n.s	n.s
*MICA*A4,*A5*	18 (2.9)	16 (3.6)	3 (9.1)	14 (4.1)	2 (1.9)	n.s	n.s	n.s	n.s	n.s	n.s
*MICA*A4,*A5.1*	58 (9.4)	33 (7.4)	2 (6.1)	28 (8.2)	5 (4.9)	n.s	n.s	n.s	n.s	n.s	n.s
*MICA*A4,*A6*	67 (10.8)	43 (9.6)	3 (9.1)	29 (8.5)	14 (13.6)	n.s	n.s	n.s	n.s	n.s	n.s
*MICA*A4,*A9*	26 (4.2)	19 (4.3)	1 (3)	16 (4.7)	3 (2.9)	n.s	n.s	n.s	n.s	n.s	n.s
*MICA*A5,*A5*	16 (2.6)	9 (2.0)	0 (0)	7 (2)	2 (1.9)	n.s	n.s	n.s	n.s	n.s	n.s
*MICA*A5,*A51*	22 (3.6)	20 (4.5)	0 (0)	18 (5.2)	2 (1.9)	n.s	n.s	n.s	n.s	n.s	n.s
*MICA*A5,*A6*	43 (7)	29 (6.5)	2 (6.1)	21 (6.1)	8 (7.8)	n.s	n.s	n.s	n.s	n.s	n.s
*MICA*A5,*A9*	23 (3.7)	19 (4.3)	1 (3)	14 (4.1)	5 (4.9)	n.s	n.s	n.s	n.s	n.s	n.s
*MICA*A51,*A51*	41 (6.6)	20 (4.5)	3 (9.1)	17 (5)	3 (2.9)	n.s	n.s	n.s	n.s	n.s	n.s
*MICA*A51,*A6*	107 (17.3)	64 (14.3)	9 (27.3)	50 (14.6)	14 (13.6)	n.s	n.s	n.s	n.s	n.s	n.s
*MICA*A51,*A9*	41 (6.6)	39 (8.7)	2 (6.1)	28 (8.2)	11 (10.7)	n.s	n.s	n.s	n.s	n.s	n.s
*MICA*A6,*A6*	77 (12.5)	63 (14.1)	3 (9.1)	45 (13.1)	18 (17.5)	n.s	n.s	n.s	n.s	n.s	n.s
*MICA*A6,*A9*	52 (8.4)	47 (10.5)	2 (6.1)	36 (10.5)	11 (10.7)	n.s	n.s	n.s	n.s	n.s	n.s
*MICA*A9,*A9*	14 (2.3)	21 (4.7)	1 (3)	16 (4.7)	5 (4.9)	n.s	n.s	0.041 (n.s)	n.s	n.s	n.s

Comparison of allele and genotype frequencies between COVID-19 patients and population reference group (PRG). P1: PRG vs. COVID-19 patients; P2: PRG vs. Asymptomatic Patients; P3: PRG vs. Moderate Patients; P4: PRG vs. Severe Patients; P5: Asymptomatic vs. Moderate Patients; P6: Asymptomatic vs. Severe Patients; Pc = *p*-value corrected by the Bonferroni test; n.s: not significant.

**Table 2 ijms-23-06979-t002:** HLA-B/MICA haplotype frequencies.

	PRG		Asymptomatic		Moderate Patients		Severe Patients		COVID-19 Patients	
*HLA-B/MICA*	F	%	F	%	F	%	F	%	F	%
**07/*A51*	105	8.5	9	13.6	58	8.5	16	7.8	83	8.8
**41/*A6*	14	1.1	2	3	10	1.5	3	1.5	15	1.6
**14/*A5*	14	1.1	2	3	8	1.2	1	0.5	11	1.2
**35/*A9*	56	4.5	2	3	38	5.5	12	5.8	51	5.4
**50/*A6*	38	3.1	1	1.5	14	2	10	4.9	25	2.6
**40/*A5*	20	1.6	1	1.5	20	2.9	3	1.5	24	2.5
**27/*A4*	41	3.3	3	4.5	14	2	8	3.9	25	2.6
**18/*A4*	125	10.1	5	7.6	58	8.5	13	6.3	75	7.9
**57/*A9*	25	2	1	1.5	22	3.2	11	5.3	34	3.6
**44/*A51*	57	4.6	4	6.1	29	4.2	10	4.9	43	4.6
**51/*A6*	102	8.3	3	4.5	39	5.7	17	8.3	58	6.1
**15/*A5*	33	2.7	3	4.5	19	2.8	9	4.4	30	3.2
**44/*A6*	124	10	8	12.1	78	11.4	29	14.1	114	12.1
**35/*A6*	20	1.6	2	3	9	1.3	2	1	13	1.4
**40/*A51*	17	1.4	2	3	8	1.2	2	1	12	1.3
**08/*A51*	53	4.3	3	4.5	36	5.2	4	1.9	42	4.4
**14/*A6*	50	4.1	2	3	37	5.4	10	4.9	49	5.2
**39/*A9*	15	1.2	1	1.5	12	1.7	2	1	16	1.7
**38/*A9*	28	2.3	1	1.5	22	3.2	10	4.9	33	3.5
**53/*A9*	15	1.2	1	1.5	15	2.2	3	1.5	19	2
**55/*A4*	9	0.7	1	1.5	13	1.9	2	1	16	1.7
**13/*A51*	19	1.5	1	1.5	9	1.3	3	1.5	13	1.4
**49/*A6*	31	2.5	1	1.5	20	2.9	7	3.4	28	3
**52/*A6*	14	1.1	1	1.5	10	1.5	5	2.4	16	1.7
**35/*A5*	41	3.3	0	0	27	3.9	6	2.9	32	3.4
**58/*A9*	10	1.06	0	0	8	1.2	2	1	16	1.3

Comparison of HLA-B/MICA haplotype frequencies between COVID-19 patients stratified by disease severity and population reference group (PRG). No significant differences were found.

**Table 3 ijms-23-06979-t003:** Characteristics and comorbidities of the study population.

Characteristics			
	Moderate Patients (*n* = 344)	Severe Patients (*n* = 106)	*p*
**Age**	63.6	61.8	n.s
**Female**	159 (46.2%)	38 (35.8%)	n.s
**Male**	185 (53.1%)	68 (64.2%)	
**UCI**	23 (6.7%)	103 (97.2%)	3.72 × 10^−74^
**No UCI**	321 (93.3%)	3 (2.8%)	
**Mechanic Ventilation**	17 (4.9%)	84 (79.2%)	1.03× 10^−50^
**No Mechanic Ventilation**	327 (95.1%)	22 (20.8%)	
**Deceased**	28 (8.1%)	41 (38.7%)	3.02 × 10^−11^
**Survivors**	316 (91.9%)	65 (61.3%)	
**Comorbidities**			
**Hypertension**	145 (42.2%)	52 (49.1%)	n.s
**DM**	65 (18.9%)	36 (34%)	0.01
**CKD**	27 (7.8%)	5 (4.7%)	n.s
**CVD**	20 (5.8%)	2 (1.9%)	n.s
**Overweight/Obesity**	41 (11.9%)	30 (28.3%)	8 × 10^−4^
**MI**	20 (5.8%)	6 (5.7%)	n.s
**HF**	16 (4.7%)	9 (8.5%)	n.s
**COPD**	20 (4.8%)	13 (12.3%)	n.s
**Asthma**	21 (6.1%)	8 (7.5%)	n.s
**PAD**	9 (2.6%)	2 (1.9%)	n.s

ICU: intensive care unit; DM: diabetes mellitus; CKD: chronic kidney disease; CVD: cerebrovascular disease MI: myocardial infarction; HF: heart failure; COPD: chronic obstructive pulmonary disease; PAD: peripheral artery disease. *p* = *p*-value; n.s: not significance.

## Data Availability

Not applicable.

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
