# Peer review of "Major Histocompatibility Complex Class I Chain-Related α (MICA) STR Polymorphisms in COVID-19 Patients"

_ijms, 2022, doi:10.3390/ijms23136979_

Round 1
Reviewer 1 Report
Thank you for considering my suggestions.
Author Response
Dear Reviewer,
Thank you for your comments and suggestions.

Reviewer 2 Report
I red about modifications and point by point response to referees.
I suggest to add point 3 as study limitation in results’ section.
Author Response
Dear Reviewer,
Thank you for your comments and suggestions.
- I suggest to add point 3 as study limitation in results’ section.
We have added this limitation in results section. (Lines 104-105)

This manuscript is a resubmission of an earlier submission. The following is a list of the peer review reports and author responses from that submission.
Round 1
Reviewer 1 Report
Thank you for the opportunity to review the manuscript Major Histocompatibility Complex Class I chain-related alpha STR Polymorphisms in COVID-19 Patients by Gutiérrez-Bautista and cols.
The manuscript shows a classical genetic association study with a MICA and HLA-B loci in 483 patients suffering from COVID-19 and a control group composed of 617 blood samples from donors from the regional blood bank of Granada. The LABType SSO typing test genotyped the HLA-B locus. At the same time, the STR polymorphism of exon 5 of MICA was performed by polymerase chain reaction (PCR) following the method of Ota (a very old method selection).
The main idea is good, and the full sample size is appropriate; however, the clinical characterization of cases is scarce. The authors fail to describe the criteria selection adequately; for example, how were asymptomatic patients identified? all of them have PCR or antigen test positive?
The same problem happens in the description for moderate and severe cases. which were the criteria to stratify in one or another group?
The major problem is the control group. Including subjects from the blood donor programs is not a great idea in infectious diseases. Were these subjects tested for SARS-CoV2 virus? Were comparable the demographical and clinical characteristics of the control group to cases?
Usually, blood donor subjects are healthy young males (mostly), and severe COVID-19 cases have a median of 62 years old; please consider realizing and regression model to adjust for potential confounders.
Author Response
Dear Reviewer,
Thank you for your value comments.
- The main idea is good, and the full sample size is appropriate; however, the clinical characterization of cases is scarce. The authors fail to describe the criteria selection adequately; for example, how were asymptomatic patients identified? all of them have PCR or antigen test positive?
The asymptomatic patients were detected among hospital workers, after being tested positive by PCR in the screenings carried out by the hospital.
All individuals included in the study had a positive PCR against SARS-CoV-2.
This has been clarified in the manuscript. Line 202 and lines 249-256.
- The same problem happens in the description for moderate and severe cases. which were the criteria to stratify in one or another group?
The criteria followed were those set by the National Institutes of Health. You can find the classification at this address:
https://www.covid19treatmentguidelines.nih.gov/overview/overview-of-covid-19/
- Coronavirus Disease 2019 (COVID-19) Treatment Guidelines. Bethesda (MD): National Institutes of Health (US); April 21, 2021.
We clarified this in the new version of the manuscript. Line 211-219.
- The major problem is the control group. Including subjects from the blood donor programs is not a great idea in infectious diseases. Were these subjects tested for SARS-CoV2 virus? Were comparable the demographical and clinical characteristics of the control group to cases?
The individuals of the control group were recruited from 2005 to 2015. Healthy individuals who did not present any disease of autoimmune etiology or immunodeficiencies were selected. If any of these individuals had developed an immune-mediated disease, it was eliminated.
This group is a representation of the allele frequencies of the area.
Other cohorts in our area have the same MICA*A9 frequency:
-Sánchez E, Torres B, Vilches JR, et al. No primary association of MICA polymorphism with systemic lupus erythematosus. Rheumatology (Oxford). 2006;45(9):1096-1100. doi:10.1093/rheumatology/kel058
- Usually, blood donor subjects are healthy young males (mostly), and severe COVID-19 cases have a median of 62 years old; please consider realizing and regression model to adjust for potential confounders.
It is known that age is a factor related to severe disease in SARS-CoV-2 infection (Gao YD, Ding M, Dong X, et al. Risk factors for severe and critically ill COVID-19 patients: A review. Allergy. 2021;76(2):428-455. doi:10.1111/all.14657).
We thank the editor for the idea for do the regression model. We did it and did not find a significant correlation between age and MICA allele frequency in the population studied. We added it in the new version of manuscript (Line 98-100).
We hope that the new version is ready to be published.

Reviewer 2 Report
Authors report their findings with MICA analysis in subjects with COVID-19
i found some issues that can be improved
-first of all pandemic raises nearly 30 months in next weeks so we learnt that variant of concerns and
vaccinations may influence the risk of mortality so they should better clarify not only the timing of data collection but also the viral variant. Otherwise they should underline this item as a great study limitation
-materials and methods should be placed before than results
-they underlined an increased rate of symptoms in presence of gene variants but it could be interesting if they could report the rate of patients that developed sepsis being a mutation that involves natural immunity
Author Response
Dear Reviewer,
Thank you for your value comments.
- First of all pandemic raises nearly 30 months in next weeks so we learnt that variant of concerns and vaccinations may influence the risk of mortality so they should better clarify not only the timing of data collection but also the viral variant. Otherwise they should underline this item as a great study limitation
The collection of samples was carried out between April 2020 and January 2021. The prevalent variant at that time in Spain was Wuhan and Alpha.
- Hodcroft EB, Zuber M, Nadeau S, et al. Emergence and spread of a SARS-CoV-2 variant through Europe in the summer of 2020. Preprint. medRxiv. 2021;2020.10.25.20219063. Published 2021 Mar 24. doi:10.1101/2020.10.25.20219063
No individual was vaccinated as no vaccines were available.
We clarified this in the new version of the manuscript. Lines 220-222.
- Materials and methods should be placed before than results.
We have used the template provide by mdpi for the generation of the manuscript. In the template the materials and methods appear after the discussion.
https://www.mdpi.com/journal/ijms/instructions
- They underlined an increased rate of symptoms in presence of gene variants but it could be interesting if they could report the rate of patients that developed sepsis being a mutation that involves natural immunity
It is a great idea to conduct this study. We have tried to carry out the study but we have not achieved a sufficiently high number of cases.
We hope that the new version is ready to be published.

Round 2
Reviewer 1 Report
Thank you for considering my comments.
As you refer, the called "control group" is not a control, this group is a representation of the allele frequencies of the area. And then should be referred to like this. I think that the only way this could be compared as a "population reference group"; in that case, this should be modified in the whole manuscript.
Another concern is the regression model done; please describe the methods employed and results in more than "p>0.05" in the adequate sections.
The clinical variables (Table 3) should show the two groups of cases in two separated columns and the corresponding statistical analysis for each comparison.
Author Response
Dear Editor,
Thank for you rapid respond.
- As you refer, the called "control group" is not a control, this group is a representation of the allele frequencies of the area. And then should be referred to like this. I think that the only way this could be compared as a "population reference group"; in that case, this should be modified in the whole manuscript.
We agree and have exchanged “control group” for “population reference group” throughout manuscript.
- Another concern is the regression model done; please describe the methods employed and results in more than "p>0.05" in the adequate sections.
We have improved the explanations of the methods used in “Results” and “Material and Methods” sections. (Line 100-104 and 284-285).
- The clinical variables (Table 3) should show the two groups of cases in two separated columns and the corresponding statistical analysis for each comparison.
We have modified the Table 3 and added the comparison between moderate and severe patient’s characteristics and comorbidities. (Line 235).
Finally, we have added a paragraph about the limitations of our work due to the age difference between population reference group and COVID-19 cases studied. (Line 202-207)
Your comments have helped make the work better.
We hope that this version is ready to be published.
Sincerely,
Juan Francisco Gutiérrez-Bautista

Round 3
Reviewer 1 Report
The authors have attended to my previous concerns, the manuscript has been improved.
Just two final suggestions:
In Table 2, remove the "HLA" in each haplotype line, and put it above in the header of the column, like this you will shorten the table length.
In the title, include the gene symbol "MICA":
Major Histocompatibility Complex Class I Chain-Related a (MICA) STR Polymorphisms in COVID-19 Patients
Author Response
Dear Editor,
Thank you for your comments.
We have made the changes you have proposed.
Sincerely,
Juan Francisco Gutiérrez-Bautista
